# Linguistic Relative Policy Optimization for Video Anomaly Reasoning

**Jiaxu Leng** [1 2]  **Jiankang Zheng** [1]  **Mengjingcheng Mo** [1]  **Zhanjie Wu** [1]  **Haosheng Chen** [1]  **Ji Gan** [1]  **Xinbo Gao** [1]

## Abstract

Video anomaly detection (VAD) with multimodal large language models has shown strong potential, yet most existing methods still depend on large-scale annotations or expert-designed priors, limiting their ability to acquire anomaly knowledge with as little human intervention as possible. To address this, we propose Linguistic Relative Policy Optimization (LRPO), which distills group-relative semantic advantages from multiple reasoning trajectories into a linguistically expressed anomaly experience prior, and adapts the model by injecting this prior into the context to steer its output distribution without any parameter updates. LRPO builds two complementary experience representations: general experience captures transferable anomaly preferences across scenarios, while scenario experience models context-dependent anomaly rules for targeted refinement. To further improve the learned experience, we introduce an anomaly alignment reward that guides trajectory optimization to match human risk preferences and reinforce temporally grounded reasoning. Extensive experiments on XD-Violence, UCF-Crime, and UBnormal demonstrate that LRPO significantly outperforms existing state-of-the-art methods under tuning-free settings.

## 1. Introduction

Video Anomaly Detection (VAD) automatically detects and localizes rare, unexpected, or hazardous events in video streams, a capability that is critical for safety-sensitive video analysis (Sultani et al., 2018; Mo et al., 2024). Despite strong performance of conventional semi-supervised and weakly supervised VAD methods (Cao et al., 2024; Zhang et al., 2024c; Wu et al., 2024; Huang et al., 2025a; Meng

---

[1]School of Computer Science and Technology, Chongqing University of Posts and Telecommunications, Chongqing, China. [2]Chongqing College of Artificial Intelligence, Chongqing, China. Correspondence to: Xinbo Gao <gaoxb@cqupt.edu.cn>.

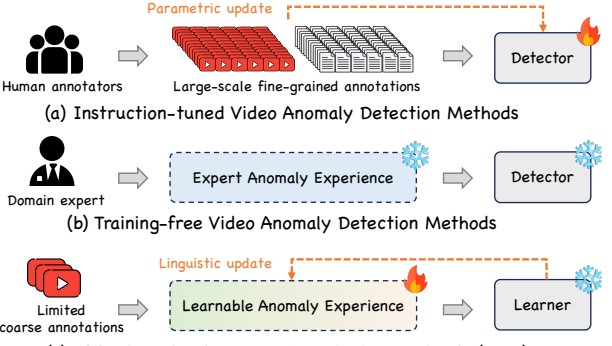

*Figure 1.* Recent research in VAD can be categorized into two types: (a) instruction-tuned VAD, which aligns anomaly knowledge by updating model parameters with large-scale fine-grained annotations; and (b) training-free VAD, which freezes model parameters and relies on expert priors to guide anomaly reasoning. We propose LRPO-based VAD: (c) it also freezes model parameters but learns linguistic anomaly experience from limited coarse annotations, substantially reducing human involvement.

et al., 2025) under the training distribution, real-world deployments are often highly diverse and commonly exhibit substantial domain shifts between training and test environments (Li et al., 2023; 2025b;a; Mo et al., 2026). Consequently, their performance typically degrades severely under out-of-distribution environments. Recently, multimodal large language models (MLLMs) have demonstrated strong generalization across a broad range of challenging visual tasks, suggesting a promising direction for enhancing the generalization of VAD under distribution shifts.

As illustrated in Figure 1, existing MLLM-based VAD methods broadly fall into two paradigms: (1) Instruction-tuned VAD. These methods (Tang et al., 2024; Zhang et al., 2025) fine-tune MLLMs to align anomaly semantics with the VAD task; however, they typically require large-scale, fine-grained annotations and thus incur non-trivial human annotation costs. (2) Training-free VAD. These methods (Zanella et al., 2024; Li et al., 2026) freeze pretrained parameters to preserve the foundation model's generalization; however, they typically depend on hand-crafted reasoning pipelines based on expert knowledge rather than data-driven feedback, hindering automated iteration and necessitating manual tuning. **Overall, both paradigms require substantial human intervention, either large-scale annotation or**

*Table 1.* Performance impact of different experience types on XD-Violence using InternVL3_5-8B as the base model.

| Experience Type | AP (%) |
| --- | --- |
| without experience | 59.93 |
| with manually designed experience | 68.47 |
| with single-trajectory learned experience | 66.78 |
| with **LRPO-learned experience (Ours)** | **73.17** |

**manual pipeline design and tuning, hindering generalizable anomaly knowledge acquisition under distribution shifts.**

By contrast, humans can often learn anomaly discrimination from limited data by leveraging broad commonsense knowledge, requiring only calibration of what constitutes an anomaly under task-specific risk preferences. Consistent with this, Table 1 compares several forms of anomaly-related experience, including human-written anomaly judgment rules and experience learned from correctness feedback on a single reasoning trajectory. The results show that directly applying a frozen vision–language model (VLM) to VAD results in poor performance, whereas providing anomaly-related experience in the context improves performance markedly. These results indicate that pretrained VLMs already encode rich general knowledge, and that the main bottleneck is the lack of explicit anomaly experience rather than limited model capacity.

Based on these observations, we propose **Linguistic Relative Policy Optimization (LRPO)**, which learns linguistically expressed anomaly experience from limited data and injects it into the model input context to improve anomaly reasoning in a tuning-free manner. A key challenge is that anomaly perception is inherently subjective and context-dependent: the same video may admit different judgments under different risk preferences, making optimization toward a single absolute label noisy and unstable. To address this, LRPO draws on relative optimization from reinforcement learning to iteratively learn and refine anomaly experience. Concretely, a *Learner* VLM generates multiple reasoning trajectories per sample to cover diverse anomaly preferences. These trajectories are then assigned reward scores, and an *Optimizer* LLM performs group-wise reflection to extract semantic advantages. The extracted advantages are distilled into anomaly experience and injected as a dynamically updated contextual signal, progressively steering the Learner's outputs toward the target risk preferences. Different from conventional reinforcement learning that optimizes model parameters, LRPO instantiates relative optimization in a language-editable experience space, where a persistent anomaly experience repository is updated by distilling semantic differences between high- and low-reward reasoning trajectories.

Within this framework, we build two complementary experience representations: **general experience**, learned by LRPO to capture transferable anomaly preferences across scenarios, and **scenario experience**, constructed by the VLM with weak-label prompting to encode context-dependent anomaly rules and expand as more data become available. Together, they support effective scenario adaptation while preserving generalization. To further improve the learned anomaly experience, we introduce an **anomaly alignment reward** with two complementary terms. The anomaly preference reward first aligns the learner's judgments with human risk preferences by using the scenario experience of the current sample as a positive reference and contrasting it with LLM-generated perturbed outputs. Building on this, the anomaly temporal dependency reward further promotes temporally grounded reasoning by contrasting performance on temporally ordered frames with randomly permuted inputs, encouraging step-by-step inference based on temporal evidence.

Our main contributions are summarized as follows: 1) We propose LRPO, a tuning-free framework for VAD that distills group-relative semantic advantages from multiple reasoning trajectories into a linguistically expressed anomaly experience prior, and adapts the model by injecting this prior into the context to steer its output distribution. 2) We build general and scenario experience to capture transferable anomaly preferences and context-dependent anomaly rules. 3) We introduce an anomaly alignment reward to optimize experience, aligning it with human risk preferences and reinforcing temporally grounded reasoning. 4) We achieve state-of-the-art tuning-free performance on XD-Violence, UCF-Crime, and UBnormal, with strong generalization across models and datasets.

## 2. Related Work

**Tuning-based VAD.** These methods update model parameters with supervision signals of varying granularity to align anomaly-related semantics with decision boundaries, including semi-supervised VAD (Yan et al., 2023; Cao et al., 2024; Zhang et al., 2024b;c; Zhu et al., 2024), weakly-supervised VAD (Sultani et al., 2018; Tian et al., 2021; Li et al., 2022; Lv et al., 2023; Shi et al., 2023; Chen et al., 2024; Wu et al., 2024; Huang et al., 2025a; Meng et al., 2025; Leng et al., 2025), and instruction-tuned VAD (Du et al., 2024; Tang et al., 2024; Zhang et al., 2024a; 2025). For example, (Rai et al., 2024) propose spatio-temporal pseudo-anomaly generation to synthesize hard cases and enhance anomaly representations and detection robustness; (Huang et al., 2025b) propose Ex-VAD, which performs explainable fine-grained anomaly detection based on vision-language models; (Zhang et al., 2025) combine an anomaly-oriented temporal sampler with an instruction-tuned MLLM

for anomaly recognition; (Huang et al., 2026) introduce Vad-R1, a representative framework that strengthens MLLM-based VAD with explicit anomaly reasoning and perception-to-cognition supervision. Although these methods typically perform strongly within the training domain, semi- and weakly-supervised methods can degrade substantially under out-of-distribution scenarios or unseen anomaly types; meanwhile, instruction-tuned methods often require large-scale fine-grained data and manually constructed instructions, incurring high human annotation costs.

**Tuning-free VAD.** These methods freeze the parameters of foundation models and directly leverage the pretrained capabilities of MLLMs for anomaly inference (Zanella et al., 2024; Yang et al., 2024; Shao et al., 2025; Cai et al., 2026; 2025b; Li et al., 2026; Lin et al., 2026; Yang et al., 2026). For example, (Zanella et al., 2024) improve anomaly scoring via cross-modal alignment while suppressing noisy descriptions; (Shao et al., 2025) segment videos into semantically coherent events and adopt hierarchical prompting for LLMs to enable anomaly localization; (Yang et al., 2026) propose an agent-based generalist VAD method that adapts to new scenes and unseen anomaly types through a closed-loop reasoning process. As they avoid domain-specific tuning, these methods largely preserve the generalization of foundation models and can be deployed quickly. However, they typically depend on hand-crafted reasoning pipelines or pre-defined agentic procedures grounded in expert knowledge rather than data-driven feedback, which hinders automated iteration and often necessitates manual tuning. Motivated by recent verbalized learning techniques (Yuksekgonul et al., 2025; Xiao et al., 2024; Cai et al., 2025a), we propose LRPO to address the above limitations. Unlike existing verbalized-learning-based VAD (Ye et al., 2025), which mainly learns guiding questions and uses them at inference time to steer the model toward local cues of anomaly patterns, LRPO explicitly models and iteratively optimizes preference experiences for anomaly judgment, thereby elevating anomaly decisions from question-guided cue focusing to experience-driven anomaly reasoning.

# 3. Method

## 3.1. Problem Formulation

Given a video $V$, video anomaly detection (VAD) aims to predict frame-level anomaly scores and localize anomalous temporal segments. Since anomalies are rare and frame-level annotations are costly, training data are typically provided with only video-level weak labels. We denote the training set as $\mathcal{D} = \{(V^{(j)}, Y^{(j)})\}_{j=1}^{N}$, where $Y^{(j)} \in \{0, 1\}$ indicates whether video $V^{(j)}$ contains anomalous events. To reduce computation while preserving key temporal evidence, we adopt the sampler in (Zhang et al., 2025) to convert each video into an $M$-frame key sequence $\tilde{V}^{(j)}$, yielding

$\tilde{\mathcal{D}} = \{(\tilde{V}^{(j)}, Y^{(j)})\}_{j=1}^{N}$. We optimize a vision-language model (VLM) on $\tilde{\mathcal{D}}$ for VAD, where the optimization target is not the model parameters but an editable linguistic anomaly experience repository. Concretely, given the sampled key-frame sequence $\tilde{V}$ and a task prompt $P$, a frozen VLM with fixed parameters $\theta_0$ induces a conditional output distribution $o \sim \pi_{\theta_0}(\cdot \mid \tilde{V}, P)$, where $o$ is a complete anomaly reasoning output. Unlike conventional paradigms that update parameters $\theta$ to directly modify $\pi_\theta$, we maintain an editable experience repository $\mathcal{E}$ and select an experience subset $\mathcal{E}(\tilde{V})$ as injected context, thereby modulating the conditional distribution to $o \sim \pi_{\theta_0}(\cdot \mid \tilde{V}, P, \mathcal{E}(\tilde{V}))$. LRPO then iteratively updates $\mathcal{E}$ to continuously steer the output distribution of $\pi_{\theta_0}$, enabling adaptation to the VAD task and aligned risk preferences without updating any VLM parameters.

## 3.2. Two Complementary Experience Representations

Anomaly reasoning requires both general reasoning principles for guidance and scene-related experience for calibrating specific anomaly decision boundaries. Therefore, LRPO adopts a two-level experience representation consisting of generic experience and scenario experience, which are maintained in an editable linguistic anomaly experience repository $\mathcal{E} = \{\mathcal{E}^{\text{gen}}, \mathcal{E}^{\text{sce}}\}$. The generic experience $\mathcal{E}^{\text{gen}} = \{e_i^{\text{gen}}\}$ is iteratively distilled by LRPO during training from multi-trajectory feedback, serving as a transferable prior for anomaly reasoning across scenes. In contrast, the scenario experience $\mathcal{E}^{\text{sce}} = \{e_i^{\text{sce}}\}$ explicitly characterizes scene-specific normal patterns and anomaly boundaries, complementing the generic prior when scene-dependent variations render the general experience insufficient. During optimization or inference, LRPO constructs an experience context $\mathcal{E}(\tilde{V})$ for an input video via a selector $\text{Sel}(\cdot)$, which injects the generic experience $\mathcal{E}^{\text{gen}}$ together with relevant scenario experience selected from $\mathcal{E}^{\text{sce}}$, thereby achieving scene adaptivity while preserving transferability. (Details of the selector are provided in §3.5.)

**Scenario Experience Construction.** Scenario experience can be constructed at scale under weak supervision. Given a training sample $(\tilde{V}^{(j)}, Y^{(j)})$, we use a predefined rule template $P_{\text{sce}}$ to prompt the frozen VLM to generate a corresponding entry, e.g., *"When in <scene type>, if <cue/event>happens (or <object>appears), you should judge it as <anomaly type>."* Here, the weak label provides the target category, i.e., the normal class or a specific anomaly category such as fire or explosion, while the VLM only supplements the scene type and visual/event cues from the sampled video to fill the remaining slots. In this way, scenario experience generation is constrained by weak supervision while avoiding manual enumeration of all combinations of scenes, cues, and anomaly categories.

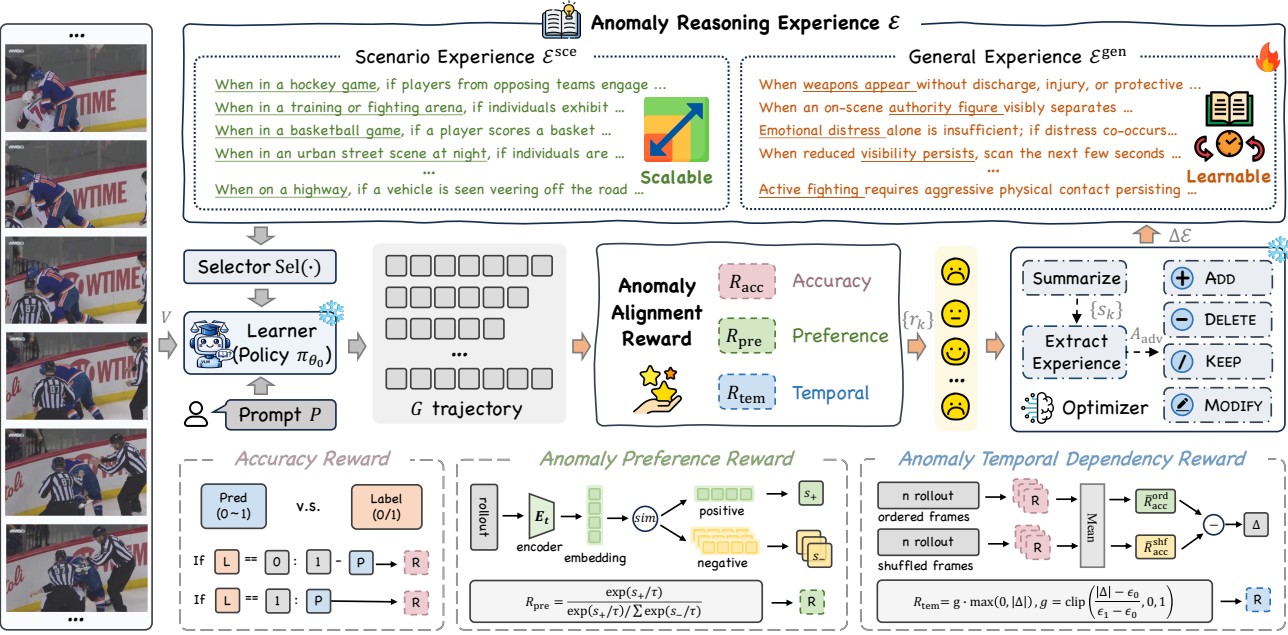

*Figure 2.* Pipeline of LRPO. LRPO leverages linguistic interaction between a *Learner* VLM and an *Optimizer* LLM to iteratively optimize the general anomaly experience under reward constraints, and injects the experience as context to adjust the Learner's output distribution, thereby adapting it to the video anomaly detection task.

Formally, we generate $e_{\text{sce}}^{(j)} = \Phi_{\text{VLM}}(\tilde{V}^{(j)}, P_{\text{sce}}, Y^{(j)})$ and write it into the repository via $\mathcal{E}^{\text{sce}} \leftarrow \mathcal{E}^{\text{sce}} \cup \{e_{\text{sce}}^{(j)}\}$. To support subsequent retrieval, we build offline indices for each scenario experience entry. Let $E_v(\cdot)$ and $E_t(\cdot)$ denote the visual and text encoders, respectively. For $e_{\text{sce}}^{(j)}$ and its associated clip $\tilde{V}^{(j)}$, we compute $\mathbf{k}_v^{(j)} = E_v(\tilde{V}^{(j)})$ and $\mathbf{k}_t^{(j)} = E_t(e_{\text{sce}}^{(j)})$, and store them in the scenario index set $\mathcal{K}^{\text{sce}} = \{(e_i^{\text{sce}}, \mathbf{k}_{v,i}, \mathbf{k}_{t,i})\}$. As data accumulate, $\mathcal{E}^{\text{sce}}$ can be incrementally expanded to cover more scenes and anomaly types.

### 3.3. LRPO Optimization Pipeline

As illustrated in Figure 2, LRPO iteratively updates linguistic anomaly experience through interactions between a learner VLM and an optimizer LLM, under the constraint of reward signals. This progressively modulates the learner's output distribution and adapts it to the VAD task. For a theoretical analysis of LRPO's optimization process, please refer to Appendix A.

**Trajectory Sampling.** To explore diverse plausible anomaly explanations and risk preferences for the same input, we perform group sampling to obtain $G$ trajectories from the learner. These parallel trajectories provide comparable candidates for reward-based ranking and subsequent semantic advantage distillation. Specifically, we denote the learner VLM as a policy $\pi_{\theta_0}$, where $\theta_0$ are fixed pretrained parameters kept frozen throughout LRPO. For the current sample

$(\tilde{V}^{(j)}, Y^{(j)})$, the selector constructs an experience context $\mathcal{E}^{(j)} = \text{Sel}(\mathcal{E}, \tilde{V}^{(j)})$ from the experience repository, and we sample a group of outputs from the conditional distribution $\pi_{\theta_0}(\cdot \mid \tilde{V}^{(j)}, P, \mathcal{E}^{(j)})$:

$$\mathcal{O}^{(j)} = \{o_k^{(j)}\}_{k=1}^G, \qquad o_k^{(j)} \sim \pi_{\theta_0}(\cdot \mid \tilde{V}^{(j)}, P, \mathcal{E}^{(j)}), \quad (1)$$

where each $o_k^{(j)}$ is a complete anomaly reasoning output.

**Reward Computation.** To constrain experience learning and provide comparable supervision signals, we evaluate each trajectory in the output group with an anomaly alignment reward $R(\cdot)$, yielding scalar rewards $r_k^{(j)} = R(\tilde{V}^{(j)}, Y^{(j)}, o_k^{(j)})$. The reward design is detailed in §3.4.

**Semantic Advantage Distillation.** Although the reward induces an ordering over sampled outputs, it is a scalar signal and does not specify *what to change* in the experience repository. We therefore use the optimizer to perform reflective group-wise comparisons and distill semantic advantages. Concretely, the optimizer first produces a structured summary for each output using a summarization template $P_{\text{sum}}$: $s_k^{(j)} = \Phi_{\text{opt}}(\tilde{V}^{(j)}, P_{\text{sum}}, o_k^{(j)})$. Given the summaries $\{s_k^{(j)}\}_{k=1}^G$, rewards $\{r_k^{(j)}\}_{k=1}^G$, and the current experience context $\mathcal{E}^{(j)}$, it contrasts high- and low-reward outputs to extract semantic advantage items with a prompt $P_{\text{adv}}$:

$$A_{\text{adv}}^{(j)} = \Phi_{\text{opt}}(\tilde{V}^{(j)}, P_{\text{adv}}, \{s_k^{(j)}, r_k^{(j)}\}_{k=1}^G, \mathcal{E}^{(j)}), \quad (2)$$

where $A_{\text{adv}}^{(j)}$ is a compact natural-language list of anomaly cues, risk boundaries, and reasoning principles associated

with higher rewards, which directly guides subsequent experience updates.

**Experience Optimization.** Given $A_{\text{adv}}^{(j)}$ and the experience repository $\mathcal{E}$, the optimizer further generates an executable edit instruction set $\Delta\mathcal{E}^{(j)}$ composed of operations such as ADD/MODIFY/DELETE/KEEP. Specifically, these operations edit individual experience entries rather than rewriting the whole repository. ADD writes a newly distilled transferable rule when the semantic advantages reveal missing anomaly cues or decision boundaries; MODIFY updates a specified entry when it is partially useful but needs correction or refinement; DELETE removes duplicated entries or entries contradicted by reward feedback; and KEEP preserves useful entries to avoid unnecessary changes. We then update the repository via $\mathcal{E} \leftarrow \text{Update}(\mathcal{E}, \Delta\mathcal{E}^{(j)})$ and proceed to the next iteration. Since experience is continuously injected as a contextual prior, the evolution of $\mathcal{E}$ directly changes the subsequent conditional distribution $\pi_{\theta_0}(\cdot \mid \tilde{V}, P, \mathcal{E})$, enabling continual adaptation without updating any model parameters. Meanwhile, the frozen learner $\pi_{\theta_0}$ serves as a strong prior that stabilizes the process and mitigates uncontrolled drift caused by experience updates.

### 3.4. Anomaly Alignment Reward

Anomaly judgment is subjective and scene-dependent, and updating experience solely from absolute supervision signals can be easily biased by incidental preferences. Meanwhile, anomaly judgment is also temporally dependent, since the same visual event may lead to different decisions under different preceding and subsequent temporal contexts. Therefore, experience optimization cannot rely only on classification correctness, but should also consider alignment with human risk preferences and reliance on temporal evidence. To this end, we design an anomaly alignment reward that evaluates trajectory quality from three aspects: *classification correctness*, *risk-preference alignment*, and *temporal-dependent reasoning*. We use it as the basis for the optimizer to conduct comparative reflection over output groups.

**Accuracy Reward.** The accuracy reward provides the most basic directional supervision. We parse an anomaly score $p(o) \in [0, 1]$ from an output $o$, and use the video-level weak label $Y \in \{0, 1\}$ as supervision:

$$R_{\text{acc}}(o) = Y \cdot p(o) + (1 - Y) \cdot (1 - p(o)). \quad (3)$$

This reward encourages higher anomaly scores when $Y = 1$ and lower anomaly scores when $Y = 0$.

**Anomaly Preference Reward.** Relying only on the accuracy reward may encourage shortcut learning based on spurious correlations in the training data, failing to align with human risk preferences. We therefore introduce an anomaly preference reward. Specifically, we take the sce-

nario experience entry $e_{\text{sce}}^{(j)}$ for the current sample as a positive preference text $e^+$, and ask an LLM to generate $H$ semantically similar but preference-mismatched perturbations $\{e_h^-\}_{h=1}^H$ as hard negatives (e.g., swapping anomaly types, removing key evidence, or altering scene conditions). Note that the positive preference text is not free-form model self-feedback; instead, it is constructed from the weak label of the current sample under a rule-based template, so its semantic direction is constrained by external supervisory signals. Let $E_t(\cdot)$ denote a text encoder. We encode the output reasoning text and preference rules as $\mathbf{z}(o) = E_t(o)$, $\mathbf{f}^+ = E_t(e^+)$, and $\mathbf{f}_h^- = E_t(e_h^-)$, and compute cosine similarities $s^+(o) = \cos(\mathbf{z}(o), \mathbf{f}^+)$ and $s_h^-(o) = \cos(\mathbf{z}(o), \mathbf{f}_h^-)$. The preference reward is defined as the softmax probability of the positive preference within the contrastive set:

$$R_{\text{pre}}(o) = \frac{\exp(s^+(o)/\tau)}{\exp(s^+(o)/\tau) + \sum_{h=1}^H \exp\big(s_h^-(o)/\tau\big)}, \quad (4)$$

where $\tau$ is a temperature hyperparameter. This reward encourages outputs to be semantically consistent with the risk preferences and anomaly cues captured by scenario experience, while staying away from perturbed preferences, thereby improving robustness of preference alignment.

**Anomaly Temporal Dependency Reward.** However, preference alignment alone may still lead the model to make anomaly decisions primarily based on static semantics, while neglecting critical temporal evidence. To encourage temporal-dependent reasoning, we introduce an anomaly temporal dependency reward. For the same video, we construct an ordered sampled sequence $\tilde{V}^{\text{ord}}$ and a shuffled sampled sequence $\tilde{V}^{\text{shf}}$, and obtain two output groups $\mathcal{O}^{\text{ord}}$ and $\mathcal{O}^{\text{shf}}$, respectively. The shuffled sequence uses the same sampled frames as the ordered sequence and only randomly permutes their temporal order. Notably, within LRPO, this ordered/shuffled comparison is used to form a reward signal for experience optimization, rather than to learn temporally sensitive video representations. Next, we compare the group-wise mean of $R_{\text{acc}}$ and define $\Delta = \bar{R}_{\text{acc}}^{\text{ord}} - \mu \cdot \bar{R}_{\text{acc}}^{\text{shf}}$, where $\mu$ controls the suppression strength for shuffled performance. We only reward improvements where the ordered input outperforms the shuffled one, and apply a soft gate to reduce noise:

$$R_{\text{tem}}^{\text{grp}} = g \cdot \max(0, \Delta), \qquad g = \text{clip}\left(\frac{|\Delta| - \epsilon_0}{\epsilon_1 - \epsilon_0}, 0, 1\right). \quad (5)$$

Finally, $R_{\text{tem}}^{\text{grp}}$ is assigned only to outputs that are correct under the ordered input (and set to 0 otherwise), ensuring that the positive advantage is meaningful. This reward measures the stable gain brought by temporal information at the group level, and encourages experience updates to rely on frame-order evidence rather than static semantics.

## 3.5. Inference with Experience Selection

**Selector.** The selector uses a scenario experience retriever $\mathrm{Ret}(\cdot)$ to return the Top-$K$ relevant entries from the scenario experience index set $\mathcal{K}^{\mathrm{sce}} = \{(e_i^{\mathrm{sce}}, \mathbf{k}_{v,i}, \mathbf{k}_{t,i})\}$ constructed in §3.2 for an input video $\tilde{V}$, and concatenates them with generic experience to form the conditional context:

$$\mathrm{Sel}(\mathcal{E}, \tilde{V}) = \mathcal{E}^{\mathrm{gen}} \oplus \mathrm{Ret}(\mathcal{K}^{\mathrm{sce}}, \tilde{V}), \qquad \left|\mathrm{Ret}(\mathcal{K}^{\mathrm{sce}}, \tilde{V})\right| = K. \tag{6}$$

Here, $\oplus$ denotes concatenation and injection under a pre-defined itemized format. Since visual-only retrieval can be confounded by cluttered backgrounds, we complement it with textual semantics that describe the scene and actions. Accordingly, $\mathrm{Ret}(\cdot)$ implements a dual-branch visual-semantic retriever. For a query video $\tilde{V}$, we construct a visual query vector $\mathbf{q}_v = E_v(\tilde{V})$ and a semantic query vector $\mathbf{q}_t$. During inference, we first let the VLM generate a video description $c = \Phi_{\mathrm{VLM}}(\tilde{V}, P_{\mathrm{cap}})$ and then set $\mathbf{q}_t = E_t(c)$; during optimization, $\mathbf{q}_t$ is directly constructed from the scenario experience text associated with the current sample. For any indexed entry $(e_i^{\mathrm{sce}}, \mathbf{k}_{v,i}, \mathbf{k}_{t,i}) \in \mathcal{K}^{\mathrm{sce}}$, we compute cosine similarities $s_{v,i} = \cos(\mathbf{q}_v, \mathbf{k}_{v,i})$ and $s_{t,i} = \cos(\mathbf{q}_t, \mathbf{k}_{t,i})$, normalize the two scores, and fuse them as $\tilde{s}_i = \alpha \, \mathrm{Norm}(s_{v,i}) + (1-\alpha) \, \mathrm{Norm}(s_{t,i})$. Finally, $\mathrm{Ret}(\cdot)$ ranks entries by $\tilde{s}_i$ and returns the Top-$K$ scenario experiences.

**Inference.** Given a test video $V$, we partition it into segments by sliding along the temporal axis with a fixed stride, where each segment $V_s$ contains $L$ consecutive frames. For each segment $V_s$, the selector constructs an experience context $\mathcal{E}(V_s) = \mathrm{Sel}(\mathcal{E}, V_s)$, and we feed $V_s$, the task prompt $P$, and $\mathcal{E}(V_s)$ into the frozen VLM (the same learner VLM used during training) to obtain the anomaly reasoning output $o_s = \Phi_{\mathrm{VLM}}(V_s, P, \mathcal{E}(V_s))$.

# 4. Experiments

## 4.1. Experimental Setup

**Datasets.** We conduct experiments on three widely used VAD benchmarks. **XD-Violence** (Wu et al., 2020) contains 4,754 untrimmed videos spanning 217 hours and covering six types of violence events collected from diverse sources, such as surveillance, movies, and online videos. We use its official split with 3,954 training videos annotated with video-level labels and 800 testing videos annotated with frame-level annotations. **UCF-Crime** (Sultani et al., 2018) consists of 1,900 untrimmed surveillance videos spanning 128 hours and covering 13 real-world anomaly categories. We follow the standard weakly supervised split, using 1,610 videos for training with video-level labels and 290 videos for testing with frame-level annotations. **UBnormal** (Acsintoae et al., 2022) is an open-set virtual dataset generated by

Cinema4D, containing 29 scenes with over 236k frames. Following the one-class setting in (Yang et al., 2024), we use the same protocol but evaluate only on its test set. Detailed normal/abnormal frame statistics for the evaluated splits are provided in Appendix B.

**Evaluation Metrics.** We adopt the standard evaluation metrics used in prior work. Specifically, we report the frame-level average precision (AP) on XD-Violence, and the area under the receiver operating characteristic curve (AUC) on UCF-Crime and UBnormal. Compared to AUC, AP is often more suitable for XD-Violence since the dataset is highly imbalanced, and AP places greater emphasis on the positive (violent) class.

**Implementation Details.** We use InternVL3_5-8B (Wang et al., 2025) as the Learner VLM and GPT-OSS-120B (Agarwal et al., 2025) as the Optimizer LLM. We train LRPO for 3 epochs. For training-subset construction, we fix the annotation budget to 100 training videos for each dataset, corresponding to 2.5% of XD-Violence and 6% of UCF-Crime due to their different training-set sizes. We randomly sample these videos uniformly across categories to reduce category imbalance. During training, we sample $M = 16$ key frames for each video and draw a group of $G = 4$ reasoning trajectories. We cap the size of the learned generic experience repository at $|\mathcal{E}^{\mathrm{gen}}| \leq 30$. For experience injection, we use CLIP4CLIP (Luo et al., 2022) as the text and visual encoders to retrieve the Top-$K$ scenario experiences with $K = 10$. During inference, we sparsely sample $L = 4$ frames per window with a temporal interval of 16 frames, and slide this window along the video stream. All experiments are conducted on two NVIDIA H800 GPUs. Additional implementation details are provided in Appendix C.

## 4.2. Main Results

**Compare with SOTA Methods.** Table 2 compares LRPO with state-of-the-art VAD methods under both tuning-based and tuning-free settings. The results show that LRPO consistently outperforms existing tuning-free baselines on all three datasets, demonstrating strong effectiveness and competitiveness. Notably, LRPO reaches 73.17% AP on XD-Violence and 85.36% AUC on UCF-Crime using only 100 training videos (corresponding to 2.5% and 6% of the training set, respectively). We observe that using the full training set yields only marginal gains (74.09% AP on XD-Violence and 86.59% AUC on UCF-Crime), suggesting that LRPO is highly sample-efficient for experience learning. Rather than fitting the data distribution via parameter updates, LRPO distills linguistically expressed anomaly experiences, making a small, class-balanced set of representative samples sufficient. Moreover, compared to VERA (Ye et al., 2025), which optimizes guiding questions to drive binary decisions, LRPO distills reusable anomaly preferences and decision

*Table 2.* Comparison with state-of-the-art methods across the XD-Violence, UCF-Crime, and UBnormal datasets.

| Methods | Venue | Supervision | Explanation | Training Data | Multi-Scenario | | Open-Set |
|---|---|---|---|---|---|---|---|
| | | | | XD-Violence / UCF-Crime | XD-Violence (AP%) | UCF-Crime (AUC%) | UBnormal (AUC%) |
| *Tuning-based Methods* | | | | | | | |
| AED-MAD (Ristea et al., 2024) | CVPR'24 | Semi | ✗ | 100% / 100% | – | – | 58.50 |
| STPAG (Rai et al., 2024) | CVPR'24 | Semi | ✗ | 100% / 100% | – | – | 57.98 |
| RFTM (Tian et al., 2021) | ICCV'21 | Weak | ✗ | 100% / 100% | 77.81 | 84.30 | 64.94 |
| PEL4VAD (Pu et al., 2024) | TIP'24 | Weak | ✗ | 100% / 100% | 85.59 | 86.76 | – |
| VadCLIP (Wu et al., 2024) | AAAI'24 | Weak | ✗ | 100% / 100% | 84.51 | 88.02 | – |
| Ex-VAD (Huang et al., 2025b) | ICML'25 | Weak | ✓ | 100% / 100% | 86.52 | 88.29 | – |
| *Tuning-free Methods* | | | | | | | |
| ZS CLIP (Radford et al., 2021) | ICML'21 | Training-free | ✓ | – | 17.83 | 53.16 | 46.20 |
| LLaVA-1.5 (Liu et al., 2024) | CVPR'24 | Training-free | ✓ | – | 50.26 | 72.84 | 53.71 |
| LAVAD (Zanella et al., 2024) | CVPR'24 | Training-free | ✓ | – | 62.01 | 80.28 | 64.23 |
| AnomalyRuler (Yang et al., 2024) | ECCV'24 | Training-free | ✓ | – | – | – | 71.90 |
| EventVAD (Shao et al., 2025) | MM'25 | Training-free | ✓ | – | 64.04 | 82.03 | – |
| URF (Lin et al., 2026) | NeurIPS'25 | Training-free | ✓ | – | 68.07 | 84.28 | 69.02 |
| VADTree (Li et al., 2026) | NeurIPS'25 | Training-free | ✓ | – | 68.85 | 84.74 | – |
| VERA (Ye et al., 2025) | CVPR'25 | Verbalized Learning | ✓ | 100% / 100% | 70.11 | 86.55 | 71.65[*] |
| **LRPO (Ours)** | – | Verbalized Learning | ✓ | 2.5% / 6% | **73.17** | **85.36** | **75.81**[*] |
| | | | ✓ | 100% / 100% | **74.09** | **86.59** | **76.24**[*] |

[*] UBnormal results are obtained by directly applying the anomaly reasoning experience (ours) or the guiding questions (VERA) learned on XD-Violence to infer on UBnormal, using InternVL3_5-8B as the backbone for fair comparison.

principles, reflecting a more cognition-driven adaptation.

**Cross-Dataset Transfer and Generalization.** As shown in Table 2, we evaluate generalization by directly transferring the anomaly reasoning experiences learned by LRPO on XD-Violence to UBnormal, without any additional training. With experience transfer alone, LRPO achieves 76.24% AUC and sets a new state of the art, suggesting that it distills reusable, language-form anomaly criteria rather than memorizing the source distribution. Under the same backbone and transfer protocol, transferring VERA's guiding questions attains only 71.65% AUC, indicating that LRPO's anomaly-preference experiences capture more transferable decision principles than question-driven cue focusing.

**Scalability and Stability Analysis.** As shown in Figure 3(a), with the general experience learned from 2.5% training data fixed, increasing the ratio of scenario experience at inference (red curve) consistently improves performance (73.17% to 74.52% AP), indicating that LRPO continues to benefit from scaling scenario experience. In contrast, scaling scenario experience during experience learning (blue curve) degrades performance (73.17% to 71.20% AP). We attribute this to shortcut learning: excessive scenario experience encourages reliance on scenario-specific cues, weakening the induction of reusable general experience and harming robustness on rare cases where matching scenario experiences are hard to retrieve. This highlights the importance of learning high-quality experience rather than simply accumulating more scenario-specific cues. To further assess stability, we independently sample five 2.5% training subsets with random seeds 42-46 to learn general experiences, and evaluate with 2.5% and 100% scenario experience at inference. As shown in Figure 3(b), the results are stable at 73.17% ± 0.51 and 74.52% ± 0.58, respectively, indicating that LRPO is robust to training-subset sampling.

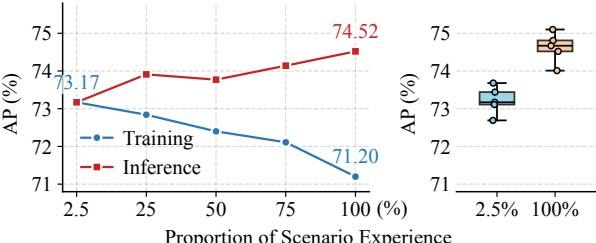

*Figure 3.* Scalability and stability of LRPO. (a) Scaling scenario experience at inference improves performance, whereas scaling it during experience learning degrades it. (b) LRPO remains stable across five random 2.5% training-subset samples.

**Effectiveness across Model Families and Scales.** Table 3 evaluates LRPO across different model families and parameter scales. LRPO delivers consistent gains with both LLaVA-NeXT-7B (Liu et al., 2024) and InternVL3_5-8B (Wang et al., 2025) as the learner (13.83% and 13.24% AP improvements), demonstrating strong cross-backbone applicability. The choice of optimizer is also flexible: using InternLM3-8B (Cai et al., 2024) improves InternVL3_5-8B from 59.93% AP to 72.77% AP, while a stronger GPT-OSS-120B (Agarwal et al., 2025) further reaches 73.17% AP, indicating that more capable reflection and experience optimization can yield additional gains. Meanwhile, this also indicates that LRPO does not rely on the strongest external model to remain effective.

### 4.3. Ablation Study

**Effect of General and Scenario Experiences.** As shown in Table 4, we analyze the effects of general experience $\mathcal{E}^{\text{gen}}$ and scenario experience $\mathcal{E}^{\text{sce}}$. The learner achieves 59.93% AP without experience injection. Injecting $\mathcal{E}^{\text{gen}}$ and learning with the accuracy reward $R_{\text{acc}}$ improves performance to 68.48% AP, suggesting that LRPO distills general ex-

*Table 3.* Effectiveness of the proposed LRPO under different learner-optimizer pairs.

| Learner | Optimizer | Method | AP (%) |
|---------|-----------|--------|--------|
| LLaVA-NeXT-7B | – | Baseline | 32.94 |
| | GPT-OSS-120B | LRPO (Ours) | 46.77 (+13.83) |
| InternVL3_5-8B | – | Baseline | 59.93 |
| | InternLM3-8B | LRPO (Ours) | 72.77 (+12.84) |
| | GPT-OSS-120B | LRPO (Ours) | **73.17** (+13.24) |

perience that effectively strengthens the learner's anomaly-aware cognition. Further adding $\mathcal{E}^{\mathrm{sce}}$ under the same setting increases performance to 70.58% AP, indicating that scenario experience complements $\mathcal{E}^{\mathrm{gen}}$ by calibrating decisions with context-specific cues.

*Table 4.* Ablation study on XD-Violence.

| Experience | | Reward | | | AP(%) |
|---|---|---|---|---|---|
| $\mathcal{E}^{\mathrm{gen}}$ | $\mathcal{E}^{\mathrm{sce}}$ | $R_{\mathrm{acc}}$ | $R_{\mathrm{pre}}$ | $R_{\mathrm{tem}}$ | |
| ✗ | ✗ | ✗ | ✗ | ✗ | 59.93 |
| ✓ | ✗ | ✓ | ✗ | ✗ | 68.48 (+8.55) |
| ✓ | ✓ | ✓ | ✗ | ✗ | 70.58 (+10.65) |
| ✓ | ✗ | ✓ | ✓ | ✗ | 69.78 (+9.85) |
| ✓ | ✗ | ✓ | ✗ | ✓ | 70.06 (+10.13) |
| ✓ | ✗ | ✓ | ✓ | ✓ | 71.91 (+11.98) |
| ✓ | ✓ | ✓ | ✓ | ✓ | **73.17** (+13.24) |

**Effect of Reward Components.** Table 4 examines how the anomaly alignment reward affects experience learning. With experience fixed to $\mathcal{E}^{\mathrm{gen}}$, using only $R_{\mathrm{acc}}$ achieves 68.48% AP. Adding anomaly preference reward $R_{\mathrm{pre}}$ improves to 69.78% AP, indicating that it discourages shortcut reasoning inconsistent with the desired anomaly preference (risk bias). Adding anomaly temporal dependency $R_{\mathrm{tem}}$ improves to 70.06% AP, suggesting that temporal-consistency constraints promote temporally grounded reasoning and yield more reliable experiences. Combining $R_{\mathrm{acc}}$, $R_{\mathrm{pre}}$, and $R_{\mathrm{tem}}$ further reaches 71.91% AP, demonstrating their complementarity. Finally, injecting $\mathcal{E}^{\mathrm{sce}}$ on top of the full reward design achieves 73.17% AP, showing that reward constraints and scenario-experience injection jointly enhance the quality and utility of learned linguistic experiences.

**Effect of Experience Optimization.** To better separate the effect of optimization from that of richer textual experience augmentation, we keep the experience injection format fixed and examine performance as the experience is progressively refined during LRPO. As shown in Table 5, AP improves steadily from 67.63% to 73.17% across the initial, one-third, two-thirds, and final stages. This suggests that the gain comes not only from adding experience, but also from continually improving its quality through optimization.

**Ablation of Scenario Experience Injection Strategy.** Keeping other settings unchanged, Table 6 ablates scenario

*Table 5.* Effect of experience optimization during LRPO.

| Experience stage | Initial | 1/3 | 2/3 | Final |
|---|---|---|---|---|
| AP (%) | 67.63 | 70.04 | 72.96 | **73.17** |

*Table 6.* Ablation on scenario experience injection strategies.

| Scenario experience injection strategy | AP (%) |
|---|---|
| No Injection | 71.91 |
| Random Injection | 71.61 (-0.30) |
| Visual-Retrieved Injection | 72.54 (+0.63) |
| **Visual-Semantic Retrieved Injection (Ours)** | **73.17** (+1.26) |

experience injection strategies. Without injection, the model reaches 71.91% AP, while random injection drops to 71.61% AP due to noisy or irrelevant experiences. Visual retrieval improves to 72.54% AP, indicating that retrieval selects scenario-matched experiences for effective contextual constraints. Our visual-semantic retrieval achieves the best 73.17% AP, showing that jointly modeling visual similarity and semantic matching yields more reliable scenario-boundary localization and experience utilization.

**Sensitivity to Generator Quality.** We further examine whether the preference reward is sensitive to generator quality. As shown in Tables 7 and 8, replacing InternVL3_5-38B with weaker InternVL3_5-14B/8B for scenario experience construction only slightly reduces AP from 73.17% to 73.02%/72.59%. Similarly, using GPT-OSS-20B instead of GPT-OSS-120B for negative sample generation still achieves 72.28% AP. These performance drops are smaller than the gains brought by the corresponding components in Table 4, indicating that generator quality mainly affects the performance ceiling, while the main improvements come from the core design of LRPO.

*Table 7.* Sensitivity of $R_{\mathrm{pre}}$ to scenario experience generator scale using InternVL3_5 (Wang et al., 2025).

| Scenario experience generator scale | 8B | 14B | 38B |
|---|---|---|---|
| AP (%) | 72.59 | 73.02 | **73.17** |

*Table 8.* Sensitivity of $R_{\mathrm{pre}}$ to negative sample generator scale using GPT-OSS (Agarwal et al., 2025).

| Negative sample generator scale | 20B | 120B |
|---|---|---|
| AP (%) | 72.28 | **73.17** |

**Sensitivity to Hyperparameters.** As shown in Figure 4, we analyze LRPO's sensitivity to key hyperparameters. (a) Increasing the rollout group size $G$ improves performance and saturates at larger $G$, indicating that more trajectories enable more stable estimation of group-relative semantic advantages and better experience distillation. (b) The general experience memory size $|\mathcal{E}^{\mathrm{gen}}|$ exhibits a "moderate-is-

best" trend: too small a memory limits coverage, whereas too large a memory may introduce redundancy and noise. (c) The Top-$K$ used for scenario experience retrieval also presents a trade-off: moderate $K$ is beneficial, while overly large $K$ leads to saturated or slightly degraded performance due to increased noise and interference. Overall, LRPO remains stable across a wide range of hyperparameters, demonstrating strong robustness.

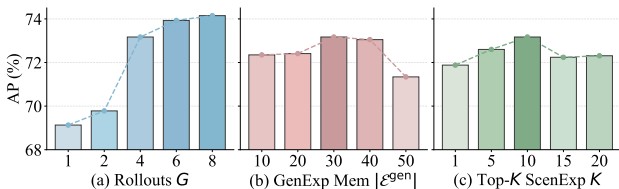

*Figure 4.* Hyperparameter ablation of LRPO. We report AP (%) when varying (a) rollout group size $G$, (b) general experience memory size $|\mathcal{E}^{\text{gen}}|$, and (c) the number of retrieved scenario experiences (Top-$K$) $K$.

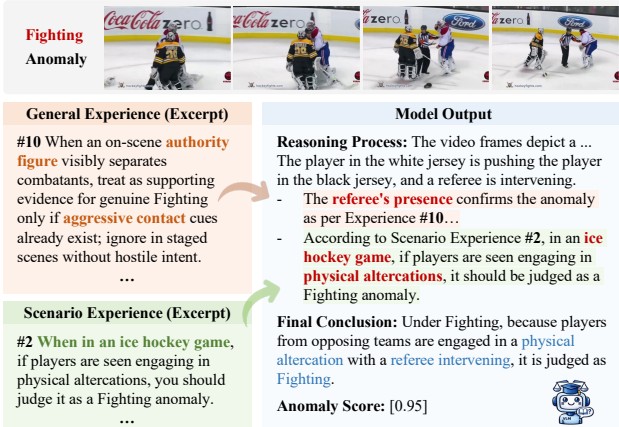

*Figure 5.* Qualitative example on video "v=251␣␣_mEwZA": by injecting learned general experience and retrieved scenario experience, the model produces anomaly preference aligned reasoning.

## 4.4. Qualitative Analysis

Figure 5 presents a fighting anomaly example to illustrate how LRPO performs interpretable anomaly reasoning by combining shared general experiences with sample-specific retrieved scenario experiences. For clarity, we show one cited general experience and one retrieved scenario experience (more experience entries are provided in Appendix D). The model cites a general experience (e.g., #10) that treats the authority figure as auxiliary evidence and requires aggressive-contact cues to confirm genuine fighting, and retrieves an ice-hockey scenario experience (e.g., #2) to calibrate the decision boundary. This example demonstrates that LRPO grounds predictions in human-readable experiences and aligns reasoning with human risk preferences, improving interpretability and decision consistency.

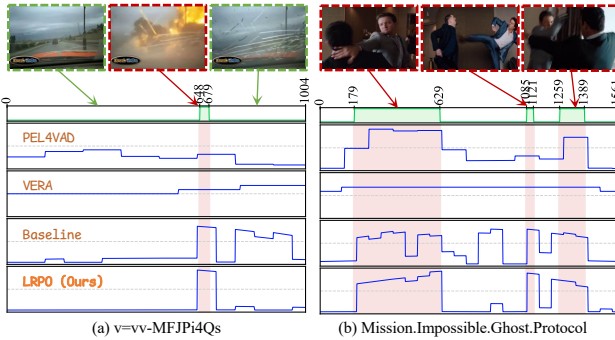

*Figure 6.* Visualization of anomaly detection results. The gray horizontal line at 0.5 indicates the anomaly detection threshold.

We visualize per-frame anomaly scores on two videos in Figure 6, comparing a tuning-based method (PEL4VAD (Pu et al., 2024)), a tuning-free method (VERA (Ye et al., 2025)), the experience-free Baseline, and LRPO. The Baseline exhibits large score fluctuations and often responds strongly on non-anomalous segments, while VERA produces smoother curves but with limited discriminability. PEL4VAD captures some anomalous events yet shows shifted boundaries or fragmented peaks. In contrast, LRPO concentrates high scores on true anomalous intervals while keeping normal segments low, indicating more consistent temporal localization and fewer false alarms. Additional qualitative analyses and visualizations are provided in the appendix E.

## 5. Limitations

Although LRPO can progressively learn and refine anomaly experience without manually enumerating all possible anomaly rules, the coverage of the learned repository remains dependent on the diversity of available training data. Rare scenes or unseen anomaly mechanisms absent from the training subset may therefore be insufficiently represented, motivating future work on automatic experience expansion and validation for open-world anomaly scenarios.

## 6. Conclusion

We propose LRPO, a novel tuning-free framework for VAD that distills group-relative semantic advantages from multiple reasoning trajectories into a linguistically expressed anomaly experience prior. To support both transferability and contextual calibration, LRPO constructs general and scenario experiences, capturing transferable anomaly preferences and context-dependent anomaly rules, respectively. We further introduce an anomaly alignment reward to optimize experiences, encouraging consistency with human risk preferences while reinforcing temporally grounded reasoning. Extensive experimental results validate the effectiveness and strong competitiveness of LRPO.

## Impact Statement

The proposed LRPO framework can support video anomaly detection in safety-critical scenarios by improving the localization and interpretation of abnormal events, such as violence, accidents, or other emergency situations. Its tuning-free and language-editable experience mechanism may also reduce the cost of adapting vision-language models to new surveillance environments.

Meanwhile, video anomaly detection systems may raise privacy, surveillance, and false-alarm concerns if deployed without appropriate safeguards. Responsible use requires lawful data collection, privacy protection, careful evaluation across diverse scenarios, and human oversight in high-stakes decision-making.

## Acknowledgements

This work was supported in part by the New Generation Artificial Intelligence-National Science and Technology Major Project under Grant No. 2025ZD0123601, in part by the National Natural Science Foundation of China under Grants No. 62472060 and 62221005, in part by the Science and Technology Innovation Key R&D Program of Chongqing under Grant No. CSTB2023TIAD-STX0016, in part by the Natural Science Foundation of Chongqing under Grants No. CSTB2024NSCQ-QCXMX0060, in part by the China Postdoctoral Science Foundation under Grant No. 2025MD774186, in part by the Chongqing Special Postdoctoral Research Funding under Grant No. 2024CQB-SHTB2002, and in part by the Chongqing University of Posts and Telecommunications Ph.D. Innovative Talents Project under Grant No. BYJS202404.

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

# A. Theoretical Analysis

Following (Brown et al., 2020), we provide a first-order analysis showing that updating linguistic experience can serve as an approximate surrogate for parameter updates. LRPO keeps the VLM parameters fixed at $\theta_0$ and modulates the conditional output distribution $\pi_{\theta_0}(o \mid C)$ by editing the experience text, where $C = [P, \tilde{V}, \mathcal{E}(\tilde{V})]$. Let $\mathbf{h}(C)$ denote the continuous representation of the context (e.g., input embeddings/hidden states). When the experience is updated from $\mathcal{E}$ to $\mathcal{E}'$, the context becomes $C' = [P, \tilde{V}, \mathcal{E}'(\tilde{V})]$, inducing a perturbation $\Delta\mathbf{h} = \mathbf{h}(C') - \mathbf{h}(C)$. Under a local linearization assumption, for any output $o$ we have the first-order approximation

$$\log \pi_{\theta_0}(o \mid C') - \log \pi_{\theta_0}(o \mid C) \approx \nabla_\mathbf{h} \log \pi_{\theta_0}(o \mid C)^\top \Delta\mathbf{h}. \tag{7}$$

Similarly, a small parameter update $\Delta\theta$ also induces a first-order change in $\log \pi_\theta(o \mid C)$. Therefore, in a first-order sense, the context perturbation $\Delta\mathbf{h}$ caused by experience editing can induce a distribution shift analogous to that of a small parameter update $\Delta\theta$, thereby changing the relative propensity of different outputs. This offers theoretical intuition for why LRPO can effectively steer the model distribution without updating any parameters.

We next interpret why LRPO can iteratively improve experience from a group-relative optimization perspective (Shao et al., 2024). For the same sample $(\tilde{V}^{(j)}, Y^{(j)})$, we draw a group of outputs $\{o_k^{(j)}\}_{k=1}^G$ under the old experience $\mathcal{E}_{\text{old}}^{(j)}$ and obtain rewards $\{r_k^{(j)}\}_{k=1}^G$, where $o_k^{(j)} \sim \pi_{\theta_0}(\cdot \mid \tilde{V}^{(j)}, P, \mathcal{E}_{\text{old}}^{(j)})$. The within-group normalized advantage is defined as $\hat{A}_k^{(j)} = \frac{r_k^{(j)} - \text{mean}(\{r_i^{(j)}\})}{\text{std}(\{r_i^{(j)}\})}$. For a candidate experience $\mathcal{E}^{(j)}$, we define the sequence-level likelihood ratio $\rho_k^{(j)} = \frac{\pi_{\theta_0}(o_k^{(j)} \mid \tilde{V}^{(j)}, P, \mathcal{E}^{(j)})}{\pi_{\theta_0}(o_k^{(j)} \mid \tilde{V}^{(j)}, \mathcal{E}_{\text{old}}^{(j)})}$, and write the group-relative objective as

$$J_{\text{LRPO}} = \mathbb{E}\left[\frac{1}{G}\sum_{k=1}^G \min\left(\rho_k^{(j)}\hat{A}_k^{(j)}, \ \text{clip}(\rho_k^{(j)}, 1-\epsilon, 1+\epsilon)\hat{A}_k^{(j)}\right) - \beta \, \text{KL}\left(\pi_{\theta_0}(\cdot \mid \tilde{V}^{(j)}, P, \mathcal{E}^{(j)}) \parallel \pi_{\theta_0}(\cdot \mid \tilde{V}^{(j)}, P)\right)\right]. \tag{8}$$

Here the clipping term and the KL regularizer jointly bound the per-step distribution change, keeping each experience update in a "small-step" regime and thus enabling stable policy improvement under the strong prior induced by the frozen base model $\pi_{\theta_0}$.

# B. Dataset Statistics

Because test videos in VAD benchmarks usually contain highly imbalanced normal and abnormal frames, quantifying this distribution is important for clarifying the experimental setting and improving evaluation transparency. Therefore, we summarize the normal/abnormal video counts and, when frame-level annotations are available, the corresponding frame distributions in Table 9. For UCF-Crime and XD-Violence, the training splits are annotated only at the video level, whereas frame-level annotations are provided for testing; therefore, their training statistics are reported only by video count. In contrast, UBnormal provides frame-level annotations for all splits, allowing frame statistics to be reported for the train, validation, and test sets.

*Table 9.* Dataset-level video counts and frame-level normal/abnormal statistics.

| Dataset | Split | # Normal Videos | # Abnormal Videos | # Total Videos | # Normal Frames | # Abnormal Frames | Abnormal Frame Ratio |
|---|---|---|---|---|---|---|---|
| UCF-Crime | Train | 800 | 810 | 1,610 | N/A | N/A | N/A |
| | Test | 150 | 140 | 290 | 1,027,477 | 84,331 | 7.59% |
| XD-Violence | Train | 2,049 | 1,905 | 3,954 | N/A | N/A | N/A |
| | Test | 300 | 500 | 800 | 1,806,004 | 529,797 | 22.68% |
| UBnormal | Train | 186 | 82 | 268 | 90,860 | 25,227 | 21.73% |
| | Validation | 26 | 38 | 64 | 14,237 | 13,938 | 49.47% |
| | Test | 53 | 158 | 211 | 42,790 | 49,850 | 53.81% |

# C. Additional Implementation Details

For the anomaly preference reward, we use InternVL3_5-38B to generate perturbed preference texts as hard negatives, with $H = 3$ per sample, and set the temperature to $\tau = 0.1$. To ensure stable and lightweight experience editing, we restrict the optimizer to perform at most 3 experience operations per update step. We cap the general experience repository at

$|\mathcal{E}^{\text{gen}}| \leq 30$ and require each retained item to describe transferable anomaly judgment principles rather than sample specific details, which keeps the repository compact and helps reduce redundancy and uncontrolled drift. For the anomaly temporal dependency reward, we set the suppression coefficient to $\mu = 0.8$ and use $\epsilon_0 = 0.05$ and $\epsilon_1 = 0.20$ for the soft gate in Eq. (5). During inference, we follow (Ye et al., 2025) to incorporate both scene context and temporal context into the reasoning process. Overall, although LRPO uses a three-stage pipeline, the workflow remains lightweight in practice, with the complete training process taking only about 2.3 hours.

## D. Experience Library Showcases

### D.1. Showcase of General Experience Learned by LRPO

We provide the complete list of general anomaly preference experiences learned by LRPO. These experiences are learned on the XD-Violence dataset using only $2.5\%$ of the training set. Each item is a natural-language rule that captures transferable decision criteria across scenarios, mainly covering temporal constraints, evidence composition, and conflict-resolution priorities, and can be directly injected as a contextual prior at inference time.

1. Active fighting requires aggressive physical contact persisting $\geq 3$ consecutive frames; weapons do not preclude Fighting if intent to harm is clear, otherwise apply Abuse when coercive handling appears without contact.

2. When weapons appear, also label Shooting if a bright or muzzle flash is followed within 3 frames by injury, distress, or protective reaction, even when the weapon is not yet visible.

3. When someone is lying on the ground, label Abuse if coercive or restraint cues appear within 2-3 frames and no weapon is visible; otherwise defer to Shooting.

4. When law-enforcement appears before civilians, label Riot if within 5 s at least two aggression cues emerge, or if a single cue persists $\geq 2$ s after the police appearance.

5. Emotional distress alone is insufficient; if distress co-occurs with discharge or protective reaction within 2-3 frames, label Shooting. If distress appears with a weapon aimed at a vulnerable person without discharge, label Abuse.

6. When multiple cues appear, label each only if its rule holds in separate segments; resolve overlaps with priority: Explosion > Shooting > Riot > Fighting > Abuse > Car-accident > Normal.

7. When restrained and another actor forcibly applies an object or substance (e.g., pours, sprays, pushes) onto the person, and visible distress follows within $\leq 3$ frames, label Abuse.

8. When reduced visibility persists, scan early, middle, and late checkpoints; if no high-severity or restraint/forced-handling cues appear and the setting looks benign, label Normal.

9. When only aftermath artifacts like lingering smoke or debris appear, no high-severity explosion cue occurred in the prior 5 seconds, and artifacts show no rapid growth, label Normal; otherwise treat as ongoing Explosion.

10. When an on-scene authority figure visibly separates combatants, treat as supporting evidence for genuine Fighting only if aggressive contact cues already exist; ignore in staged scenes without hostile intent.

11. When a crowd shifts from passive to aggressive, label Riot if two of: density rise, chanting, aggression cues (projectiles, tear-gas, protest symbols, fire, smoke, flares) appear within 5 seconds; cue persisting $\geq 1$ frame.

12. When a vulnerable individual is restrained by a dominant aggressor and visible distress appears, label Abuse; weapon presence does not downgrade unless a discharge flash or weapon-caused injury occurs within two frames.

13. When a sudden bright flash is large and sustained, and within the next few frames smoke, fire, debris, shockwave, or damage appear, prioritize Explosion over Shooting, even if a weapon is present.

14. When reduced visibility persists, scan the next few seconds for abrupt motion, sudden light or sound cues, emergency vehicles, or rapid crowd shifts; if detected, invoke the matching specific anomaly rule.

15. When weapons appear but no discharge cue in 2-3 frames, run the flash-explosion scan; if Explosion is detected, label Explosion, else do not label Shooting and monitor or consider Abuse.

16. When a sudden bright flash is detected, scan the next five frames for explosion cues (smoke, debris, fire, shockwave, damage); if found, label Explosion, overriding other cues.

17. When law-enforcement and civilians confront and officers wear riot gear, treat megaphones, banners, chanting, raised fists, coordinated gestures, or density surge as Riot cues persisting $\geq 1$ frame.

18. When a vulnerable individual shows visible injury and a dominant aggressor is present, require a coercive action (e.g., grip, push) within ≤2 frames followed by distress within ≤3 frames to label Abuse.

19. When a sudden bright flash occurs without explosion cues, label Shooting only if a weapon-related cue appears or injury/distress directly follows the flash within 2-3 frames; otherwise continue scanning.

20. When a sudden bright flash (e.g., flare or fireworks) occurs with law-enforcement and a hostile crowd, and any aggression cue appears within 3 seconds, label Riot with elevated confidence.

21. When an anomaly cue appears and any label is assigned, always continue scanning remaining frames for independent cues of other categories, applying each rule and resolving overlaps by priority.

22. When a vehicle abruptly changes motion, and within five seconds any of: damage, smoke, fire, debris, emergency-response lights, or clear traffic disruption (stopped vehicles, congestion, lane blockage) appear, label Car-accident, overriding lower-severity cues.

23. When weapons appear, prioritize Shooting over Fighting unless the weapon is sport equipment in a recognized sport context; then evaluate aggressive contact under the Fighting rule if no discharge or injury cues.

24. When weapons appear without discharge, injury, or protective reaction in 3-5 frames, prioritize Abuse labeling if restraint or distress cues exist; only label Shooting with a discharge flash or direct weapon-injury link.

25. When an anomaly cue appears and no high-severity cues are detected across early, middle, and late segments, and behavior is ordinary for the setting, assign the Normal label.

26. When a sudden bright flash is large and smoke plumes, flames, or fireballs persist for three or more frames, especially with shockwave or debris, label Explosion regardless of weapon.

27. When law-enforcement appears before civilians wearing riot gear, label Riot if two aggression cues span ≥3 seconds, boosting confidence as strong temporal evidence.

## D.2. Showcase of Selected Scenario Experience

We provide an excerpt of the scenario experience entries. Each entry is a natural-language rule that specifies context-dependent anomaly cues, triggering conditions, and decision boundaries, and can be retrieved and injected as an in-context prior during inference.

- **Explosion.**

  - When in an urban combat scene, if a person aims and fires a weapon from a rooftop, causing an explosion inside a building, you should judge it as a shooting and explosion anomaly.
  - When in a combat scene, if gunfire and explosions are visible, you should judge it as a shooting and explosion anomaly.
  - When in a battlefield scene, if soldiers are seen aiming and firing weapons at surrendering individuals, and there are signs of explosions and fire, you should judge it as a shooting and explosion anomaly.
  - When in an industrial or high-tension scene, if an explosion occurs (fire, smoke, debris), you should judge it as an explosion anomaly.

- **Shooting.**

  - When in a street scene, if a person points a gun at another person and fires, you should judge it as a shooting anomaly.
  - When in an indoor scene, if a person is holding a gun and a muzzle flash is visible, you should judge it as a shooting anomaly.
  - When in a combat scene, if people are aiming rifles and firing, you should judge it as a shooting anomaly.
  - When in a robbery-like scene, if a weapon is brandished and a shot is fired, you should judge it as a shooting anomaly.

- **Riot.**

  - When in a street protest scene, if a crowd confronts law enforcement and aggressive actions occur, you should judge it as a riot anomaly.

- When in a public disturbance scene, if people are throwing objects or clashing with police, you should judge it as a riot anomaly.
- When in an outdoor crowd scene, if a mob becomes violent and pushes or attacks, you should judge it as a riot anomaly.
- When in a protest scene, if tear gas, shields, or riot gear appears with hostile crowd behavior, you should judge it as a riot anomaly.

- **Fighting.**

  - When in a street scene, if two or more people are punching or kicking each other, you should judge it as a fighting anomaly.
  - When in an indoor scene, if a physical fight breaks out with sustained aggressive contact, you should judge it as a fighting anomaly.
  - When in a public place, if multiple individuals engage in a brawl, you should judge it as a fighting anomaly.
  - When in a hallway scene, if people grapple and strike each other, you should judge it as a fighting anomaly.

- **Abuse.**

  - When in a domestic scene, if a person forcibly restrains or drags another person and distress is visible, you should judge it as an abuse anomaly.
  - When in an indoor scene, if a vulnerable individual is restrained and coerced, you should judge it as an abuse anomaly.
  - When in a street scene, if an aggressor holds down a person and applies coercive force, you should judge it as an abuse anomaly.
  - When in a room scene, if a person is being tied up or held captive, you should judge it as an abuse anomaly.

- **Car accident.**

  - When in a road scene, if vehicles collide and damage is visible, you should judge it as a car accident anomaly.
  - When in a highway scene, if a vehicle crashes and smoke or debris appears, you should judge it as a car accident anomaly.
  - When in an intersection, if a sudden collision causes traffic disruption, you should judge it as a car accident anomaly.
  - When in a street scene, if a car hits an object or another vehicle and people react, you should judge it as a car accident anomaly.

- **Normal.**

  - When in a sports competition scene, if the actions are consistent with play and no harm cues appear, you should judge it as normal.
  - When in a stage performance scene, if violence-like gestures appear but are clearly choreographed, you should judge it as normal.
  - When in a training or drill scene, if weapons appear without hostile intent or discharge cues, you should judge it as normal.
  - When in a construction scene, if sparks or bright lights occur as part of work activity, you should judge it as normal.

## E. Additional qualitative analyses

Figure 7 provides a normal-case example illustrating how LRPO combines learned general experience with retrieved scenario experience to support preference-aligned reasoning. The general experience items (e.g., #22 and #25) act as transferable decision criteria, prompting the model to verify whether abrupt motion changes, high-severity cues, or other accident evidence emerge across temporally separated segments. Meanwhile, the retrieved scenario experience (e.g., #1) supplies a context-dependent trigger rule for highway driving, calibrating what constitutes a car-accident boundary under this scene. Since the frames exhibit orderly driving and lack the required evidence (no abrupt motion, disruption, or damage cues),

the model consistently rules out anomalies and outputs a normal conclusion with a low anomaly score. Overall, this case suggests that general experience provides reusable anomaly preferences and conflict-resolution principles, while scenario experience anchors these preferences to scene-specific boundaries, jointly improving reasoning consistency.

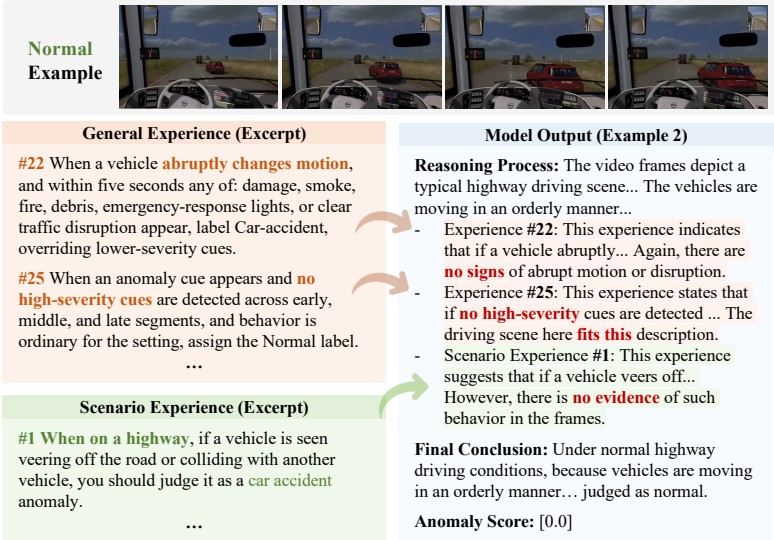

*Figure 7.* Qualitative example on video "v=y7JEq-kf2I". General experience provides transferable criteria (e.g., checking abrupt motion and high-severity cues), while scenario experience supplies scene-specific triggers and boundary calibration for the highway-driving context, leading to a consistent normal conclusion.

Figure 8 visualizes several frame-level anomaly score curves on the XD-Violence dataset. We compare LRPO with representative tuning-based methods, including HyperVD (Zhou et al., 2024), MACIL_SD (Yu et al., 2022), UR-DMU (Zhou et al., 2023), PEL4VAD (Pu et al., 2024), and DSRL (Leng et al., 2024), as well as tuning-free methods such as LAVAD (Zanella et al., 2024) and VERA (Ye et al., 2025). Across these examples, LRPO produces score trajectories that better align with the temporal extent of ground-truth anomalous intervals: it yields clearer separation between normal and abnormal regions, assigns higher confidence within anomalous segments while suppressing spurious spikes in normal frames, and exhibits more stable temporal consistency near boundaries. These qualitative results suggest that injecting and iteratively optimizing anomaly experience helps LRPO form more reliable temporally grounded anomaly evidence, complementing both training-based detectors and existing tuning-free approaches.

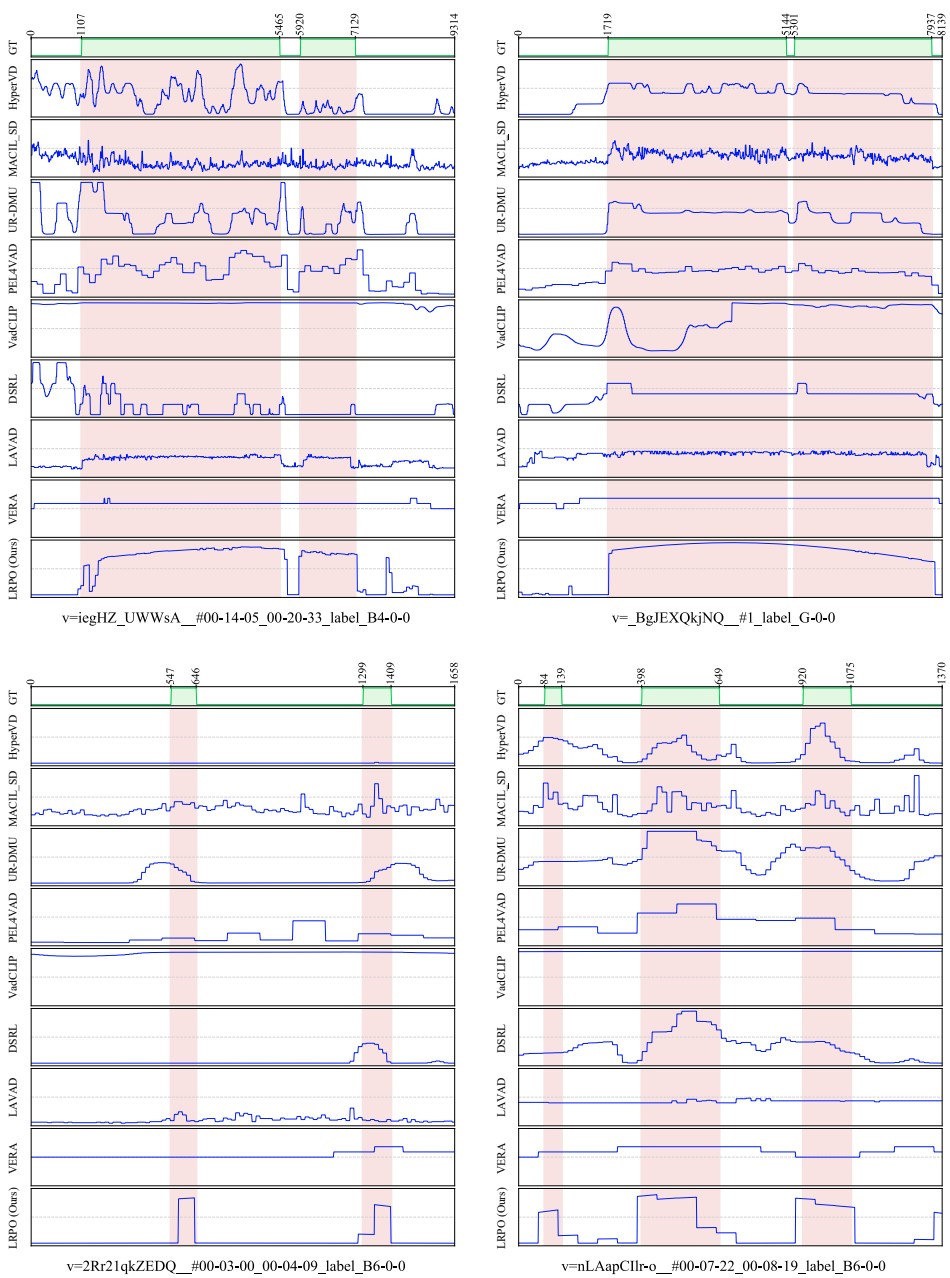

*Figure 8.* Additional visualization of frame-level anomaly detection results on XD-Violence. The gray horizontal line at 0.5 indicates the anomaly detection threshold.

