# OpenReview forum: "Linguistic Relative Policy Optimization for Video Anomaly Reasoning"
_ICML.cc/2026/Conference — ICML 2026 regular_

### Official Review · Reviewer_hZ1U · 2026-03-09

**Soundness:** 3
**Presentation:** 2
**Significance:** 4
**Originality:** 3
**Overall Recommendation:** 4
**Confidence:** 2

**Summary:**

This paper introduces Linguistic Relative Policy Optimization (LRPO) for video anomaly detection using multimodal large language models. The method distills group-relative semantic advantages from multiple reasoning trajectories into a linguistically expressed anomaly experience prior, which is injected into the model’s context to guide its output distribution. It further models both general experience and scenario-specific experience to capture transferable anomaly preferences as well as context-dependent anomaly rules. In addition, the approach introduces an anomaly alignment reward that optimizes the learned experience by aligning it with human risk preferences while reinforcing temporally grounded reasoning. Experiments demonstrate state-of-the-art tuning-free performance across several video anomaly detection benchmarks.

**Compliance With Llm Reviewing Policy:**

Affirmed.

**Final Justification:**

I think the rebuttal has addressed my concern so I raised the score.

**Key Questions For Authors:**

Because I may not be an expert in this specific area, some of my confusion could be due to misunderstanding. It would be helpful if the authors could clarify the following points:

1) Explanation of Table 1. When discussing manually designed experience and single-trajectory learned experience, the paper refers to Table 1 but does not clearly explain these concepts before using them. This makes the table difficult to interpret.

2) Connection between the figure and the method. The overview figure is visually clear, but it would be more helpful if it explicitly used the same symbols as the mathematical formulation so that readers can more easily map the figure to the method.

3) Notation simplification. Are all introduced symbols necessary? Some notation (e.g., around lines ~142 and ~152 for the video representation) seems redundant and could potentially be simplified.

4) the author mentioned the template .“When in<scene type>, if <cue/event>happens(or<object>appears),you should judge it as<anomaly type>”.  I am wondering how to list all the possible pairs.

5) In the Anomaly Temporal Dependency Reward, it looks like borrowing idea from video self-supervised learning shuffle and learn. Besides I don’t get why we need to encourages experience updates to rely on frame-order evidence rather than static semantics?

6) is 154 experience prior token?

At the current stage, I lean toward weak reject for two main reasons: The paper is difficult to read and follow, which makes it challenging to fully evaluate the method; I am uncertain about the degree of novelty relative to existing work. That said, the experimental evaluation is extensive and generally supports the claims of the paper. If the authors can clarify the presentation issues and better articulate the novelty, I would be open to revising my score, especially if other reviewers view the contribution more positively.

**Limitations:**

The authors do not provide limitations of the paper. I think the authors should discuss it, e,g. if humans need to list all potential anomaly for the methods to work?

**Strengths And Weaknesses:**

Strengths

Presentation. The paper provides a clear motivation and includes a well-designed overview figure that helps convey the high-level idea. However, several parts of the text are difficult to follow, and some explanations appear later than when the concepts are first introduced, which makes the method harder to understand.

Soundness. The proposed method appears technically sound, and the paper presents extensive experimental results across multiple benchmarks to support its claims.

Significance. Video anomaly detection is an important and challenging problem, and improving multimodal reasoning for this task could have meaningful impact.

Originality. The core idea appears interesting and potentially novel. However, since this is not my primary research area, I am not entirely confident in assessing the level of novelty relative to existing work.

Weaknesses

Clarity and readability. The paper is difficult to read in several places. Some concepts are referenced before they are clearly defined, and explanations are occasionally delayed until later sections, which makes the technical narrative harder to follow.

Notation complexity. The paper introduces many symbols, some of which appear unnecessary and increase the cognitive load for readers (e.g., notation for video in lines ~142 and ~152). Simplifying the notation could improve clarity.

Limited explanation of certain design choices. Some components of the method are introduced with limited intuition or justification, which makes it harder to understand the motivation behind them.

Novelty uncertainty. While the idea appears interesting, it is not entirely clear how strongly it differs from existing approaches in related areas such as reward design or reasoning trajectory aggregation.

---

> ### Author Rebuttal · Authors · 2026-03-31
>
> We sincerely thank the reviewer for the thoughtful evaluation of our work, including the clear motivation and overview figure (`provides a clear motivation and includes a well-designed overview figure`), the technical soundness (`the proposed method appears technically sound`), and the extensive experimental support (`the paper presents extensive experimental results across multiple benchmarks`). We also appreciate the concerns on clarity, notation, and novelty articulation, and address them below. For clarity, we use "W#", "Q#", and "L#" for weaknesses, questions, and limitations.
> > **W1/Q1: On clarity and Table 1**
>
> We agree that some concepts should be introduced earlier. In the revision, we will define them before Table 1 is first referenced and clarify them in the caption. Specifically, manually designed experience refers to human-written anomaly judgment rules, while single-trajectory learned experience refers to experience learned only from correctness feedback on a single reasoning trajectory.
> > **W2/Q2/Q3: On notation and figure-method connection**
>
> We agree that the notation can be simplified. In the revision, we will remove redundant symbols (e.g., for video representation), reduce dense superscripts/subscripts, and align the overview figure with the formulation using the same core symbols.
> > **W3: On design motivation**
>
> We appreciate this comment and provide a clearer explanation of our design motivation below. The motivation for the **two experience representations** is that anomaly reasoning needs both general reasoning principles for guidance and scene-related experience for calibrating specific anomaly decision boundaries. As for the **anomaly alignment reward**, experience optimization cannot rely only on classification correctness, because anomaly itself is relative and anomaly judgment is temporally dependent; it must also be constrained by whether it aligns with human risk preferences and whether it genuinely relies on temporal evidence.
> > **W4: On novelty vs. reward design / trajectory aggregation**
>
> We agree that these distinctions should be articulated more clearly. Unlike rewards mainly designed to improve answer correctness or formatting, **LRPO's reward is designed to optimize anomaly experience for VAD.** Because anomaly judgment is both relative and temporally dependent, LRPO goes beyond correctness by additionally modeling risk-preference alignment and reliance on temporal evidence, which bring +1.85 and +1.58 AP in Table 4.
> Likewise, **LRPO is not simply trajectory aggregation by score.** In standard policy optimization, reward differences are converted into numerical advantages for updating model parameters. In contrast, LRPO compares the semantic differences between high- and low-reward trajectories, aggregates them into relative semantic advantages, and distills them into experience updates. This avoids VLM parameter tuning while making the optimization process more interpretable, and the learned experience remains editable and reusable.
> > **Q4/L1: On template coverage and limitations**
>
> We appreciate the reviewer’s concern. LRPO does not require manually enumerating all possible <scene, cue/event, anomaly> pairs or potential anomalies in advance. Instead, it **follows a data-sample-driven process that progressively accumulates and refines anomaly experience**. For each training sample, under weak-label supervision, the frozen VLM is prompted with a template to generate sample-specific scenario experience from the current video content, which is then gradually written into the scenario experience repository. A practical limitation is that the method depends on the coverage and diversity of the training data, and we will make this point more explicit in the revised paper.
> > **Q5: Is the temporal dependency reward inspired by Shuffle-and-Learn?**
>
> We appreciate this question. Our temporal dependency reward is related in intuition to Shuffle-and-Learn, since both use ordered/shuffled comparison to probe temporal information. However, this construction serves different roles in different frameworks. Shuffle-and-Learn uses it as a self-supervised objective to learn temporally sensitive video representations, whereas **we use the ordered/shuffled reasoning gap as a reward signal to assess and guide whether anomaly reasoning truly relies on temporal evidence, thereby serving experience optimization**. This matters because static semantics alone may encourage appearance shortcuts and miss dynamics that often determine anomaly, such as the temporal progression underlying falling, collision, or violence. By doing so, the temporal reward encourages experience updates to focus more on temporal cues relevant to anomaly judgment, and its effectiveness is supported by the 1.58 AP gain in Table 4.
> > **Q6: On "experience prior"**
>
> We apologize for the unclear wording. Experience prior refers to anomaly experience in natural language, not to a soft prompt token or special token embedding.

---

> > ### Author Rebuttal · Reviewer_hZ1U · 2026-04-01
> >
> > Thanks authors for the response. My concerns are addressed.

---

> > > ### Author Response · Authors · 2026-04-02
> > >
> > > Thank you for your thoughtful follow-up and for recognizing our rebuttal. We sincerely appreciate your time, constructive feedback, and positive update.

---

### Official Review · Reviewer_bW7Z · 2026-03-10

**Soundness:** 3
**Presentation:** 3
**Significance:** 4
**Originality:** 3
**Overall Recommendation:** 4
**Confidence:** 3

**Summary:**

A multimodal large language model–based video anomaly detection (VAD) framework is proposed through a Linguistic Relative Policy Optimization (LRPO) strategy. The proposed approach introduces a novel VAD framework, termed LRPO, designed to leverage semantic reasoning derived from language-guided inference.
The framework extracts collective relative semantic advantages from multiple reasoning trajectories and transforms them into language-expressed anomaly experience priors. These priors are subsequently injected into contextual prompts to steer the model’s inference behavior, thereby guiding the output distribution without requiring any parameter updates. In this manner, anomaly detection is achieved through language-driven reasoning rather than conventional model retraining.
Within this framework, general experience captures cross-scene anomaly preferences that exhibit strong transferability across heterogeneous environments, whereas scene-specific experience models context-dependent anomaly rules to enable targeted optimization under particular surveillance scenarios. Such a dual-experience design allows the model to balance global generalization and local contextual sensitivity.
To further enhance the quality of learned experience, an anomaly alignment reward is introduced. This reward mechanism guides trajectory optimization toward patterns that better align with human risk preferences, while simultaneously reinforcing temporal reasoning within video understanding. As a result, the model becomes more capable of capturing temporally evolving abnormal behaviors and producing semantically coherent anomaly judgments.
Extensive experiments conducted on XD-Violence, UCF-Crime, and UBNormal demonstrate that, under a tuning-free setting, the proposed LRPO framework consistently outperforms existing state-of-the-art approaches. The results highlight the effectiveness of language-guided experience extraction and contextual reasoning for robust and transferable video anomaly detection.

**Compliance With Llm Reviewing Policy:**

Affirmed.

**Key Questions For Authors:**

Not available

**Limitations:**

Not available

**Strengths And Weaknesses:**

Strengths
1.	The ablation study demonstrates a substantial performance improvement achieved by the LRPO framework, indicating that the proposed design effectively contributes to enhanced anomaly detection capability.
2.	The analysis presented in Effectiveness of the Proposed LRPO under Different Learner–Optimizer Pairs further shows that LRPO consistently yields significant performance gains across various learner–optimizer combinations, highlighting the robustness and adaptability of the proposed approach.
Weaknesses
3.	The distribution ratio between normal and abnormal frames in the testing datasets is not clearly specified. Since video anomaly detection benchmarks are typically highly imbalanced, a quantitative description of the imbalance level would help clarify the experimental setting and improve the transparency of the evaluation.
4.	In general research paper writing, the frequent use of “we” as the grammatical subject is often discouraged. A more formal academic style is typically achieved by adopting an objective tone or passive constructions to enhance clarity and scholarly presentation.

---

> ### Author Rebuttal · Authors · 2026-03-31
>
> We sincerely thank the reviewer for the encouraging feedback on our work, including the acknowledgment that LRPO brings clear gains in the ablation study (`the ablation study demonstrates a substantial performance improvement achieved by the LRPO framework`), as well as the observation that LRPO remains effective across different learner–optimizer pairings, which supports the robustness and flexibility of the proposed framework (`LRPO consistently yields significant performance gains across various learner–optimizer combinations`). We have carefully considered the reviewer’s comments and provide detailed responses below. For clarity, we use "W#" to denote weakness comments.
>
> > **W1: The distribution ratio between normal and abnormal frames in the testing datasets is not clearly specified. Since video anomaly detection benchmarks are typically highly imbalanced, a quantitative description of the imbalance level would help clarify the experimental setting and improve the transparency of the evaluation.**
>
> We sincerely thank the reviewer for this helpful suggestion. We agree that quantitatively describing the normal/abnormal distribution would improve the clarity and transparency of the evaluation setting. We therefore additionally computed the numbers of normal and abnormal frames, together with the abnormal frame ratio, for all splits where frame-level annotations are available.
>
> Specifically, the abnormal frame ratio is 7.59% on the UCF-Crime test set, indicating a highly imbalanced distribution; 22.68% on the XD-Violence test set, which is also notably imbalanced; and 53.81% on the UBnormal test set, which is relatively more balanced. These statistics help contextualize the evaluation settings across benchmarks and also explain metric choice: AP is particularly informative on XD-Violence, where positive-class performance is more critical, whereas AUC is reported on UCF-Crime and UBnormal following common practice.
>
> It is also worth noting that UCF-Crime and XD-Violence are weakly supervised VAD datasets whose training splits provide only video-level labels rather than frame-level annotations. Therefore, their training-set frame ratios cannot be computed accurately. In contrast, UBnormal provides frame-level annotations for all splits, so we report its frame-level statistics for training, validation, and testing. Accordingly, we retain video-level statistics for the training splits of UCF-Crime and XD-Violence, while reporting frame-level distributions wherever such annotations are available.
>
> |Dataset|Split|# Normal Videos|# Abnormal Videos|# Total Videos|# Normal Frames|# Abnormal Frames|Abnormal Frame Ratio|
> |:-:|:-:|:-:|:-:|:-:|:-:|:-:|:-:|
> |UCF-Crime|Train|800|810|1610|-|-|-|
> |UCF-Crime|Test|150|140|290|1,027,477|84,331|7.59%|
> |XD-Violence|Train|2049|1905|3954|-|-|-|
> |XD-Violence|Test|300|500|800|1,806,004|529,797|22.68%|
> |UBnormal|Train|186|82|268|90,860|25,227|21.73%|
> |UBnormal|Validation|26|38|64|14,237|13,938|49.47%|
> |UBnormal|Test|53|158|211|42,790|49,850|53.81%|
>
> > **W2: In general research paper writing, the frequent use of "we" as the grammatical subject is often discouraged. A more formal academic style is typically achieved by adopting an objective tone or passive constructions to enhance clarity and scholarly presentation.**
>
> Thank you for this careful suggestion. We agree that the current manuscript still has room for improvement in writing style, and that frequent use of first-person subjects may weaken the objectivity and formality of academic presentation. While this issue does not affect the method, experiments, or conclusions themselves, it does affect the overall polish of the paper. In the final version, we will systematically review the manuscript and reduce unnecessary uses of "we" as the subject wherever possible, rewriting them into more objective and formal academic expressions without sacrificing readability.

---

> > ### Author Rebuttal · Reviewer_bW7Z · 2026-04-03
> >
> > Thank you to the authors' rebuttal for pointing out the two main weaknesses (W1 and W2).

---

> > > ### Author Response · Authors · 2026-04-03
> > >
> > > Thank you for your constructive feedback and for updating your review. We are pleased that our rebuttal has addressed your concerns.

---

### Official Review · Reviewer_DTt8 · 2026-03-11

**Soundness:** 3
**Presentation:** 3
**Significance:** 3
**Originality:** 3
**Overall Recommendation:** 5
**Confidence:** 5

**Summary:**

Unlike prior methods that often rely on substantial manual intervention, the authors propose LRPO, a tuning-free framework for video anomaly reasoning. It learns linguistically expressed anomaly experience from limited data and injects it into the model input context to improve anomaly reasoning. LRPO constructs two types of experience representations, namely general experience and scenario experience, to characterize transferable anomaly risk preferences and scenario-dependent anomaly rules, respectively. In addition, it designs an anomaly alignment reward to optimize the experience bank, making it better aligned with human risk preferences and strengthening temporally grounded reasoning.

**Compliance With Llm Reviewing Policy:**

Affirmed.

**Final Justification:**

The authors have addressed all my concerns, thus I maitain my positive score.

**Key Questions For Authors:**

Please see Weakness.

**Limitations:**

Please see Weakness.

**Strengths And Weaknesses:**

Strength
1) The idea is novel. It is the first to use relative optimization to learn linguistically expressed anomaly experience, enabling genuine experience-based anomaly reasoning rather than superficial correlation-based classification.
2) The method is generalizable. The general and scenario experience design captures both transferable anomaly preferences and scenario-specific anomaly rules, enabling robust cross-scenario generalization.
3) The experiments are comprehensive and the gains are consistent, yielding the strongest results under the tuning-free setting across multiple benchmarks.

Weakness
1) The implementation details of the experience bank editing mechanism remain unclear, such as the concrete procedure of ADD/DELETE/MODIFY/KEEP, and the number of edited entries and associated constraints per iteration. Further clarification would be helpful.
2) The authors use 2.5% and 6% of the training data for few-shot training on XD-Violence and UCF-Crime, respectively, but the rationale for choosing different sampling ratios across the two datasets is not sufficiently explained. Moreover, the construction protocol of the few-shot subsets is also unclear, and further discussion is needed.
3) The construction of the shuffled variant in the anomaly temporal dependency reward is not sufficiently clear, such as the shuffling granularity and whether the same set of frames is preserved. Further clarification would improve reproducibility.
4) There is minor room for improvement in writing and formatting. Some symbol definitions and pipeline descriptions are relatively dense, and the final version could be further refined to improve readability. In addition, there is a minor formatting issue where the paper writes “As illustrated in Figure. 2”, which should be “Figure 2”.

---

> ### Author Rebuttal · Authors · 2026-03-31
>
> We sincerely thank the reviewer for the positive assessment of our work, including the recognition of the method’s novelty (`the idea is novel`), its generalizability (`the method is generalizable`), and the comprehensiveness and consistency of the experimental results (`the experiments are comprehensive and the gains are consistent`). We have carefully considered the reviewer’s constructive comments and provide detailed responses below. For clarity, we use "W#" to denote weakness comments.
>
> > **W1: The implementation details of the experience editing mechanism are not sufficiently clear.**
> We appreciate this question and agree that the implementation details should be made clearer. The experience bank editing mechanism is not an unconstrained free-form rewriting process. Instead, the optimizer generates a constrained set of edit operations based on the current experience bank and the extracted semantic advantages. Specifically, in each iteration, the optimizer first reads the current general experience together with the semantic advantages distilled from comparisons between high- and low-reward trajectories, and then selects only a small number of necessary edits from three operation types: ADD, UPDATE, and DELETE. Here, UPDATE replaces an existing experience entry, ADD appends a new experience entry to the end of the bank, and DELETE is used cautiously only when an entry is clearly redundant or misleading. KEEP corresponds to leaving an entry unchanged, and therefore does not necessarily need to be written as an explicit editing command in implementation. To ensure stability, we impose explicit constraints on each iteration: at most 3 edit operations are allowed per iteration, and the total number of general experience entries is capped at 30. In addition, each experience entry must satisfy several formatting and content constraints, such as limited length, emphasizing transferable reasoning patterns, avoiding operational details, highlighting robust decision points, and minimizing conceptual redundancy across entries. We will clarify this editing procedure and its constraints in the revised version so that the execution of ADD / DELETE / MODIFY / KEEP becomes more transparent.
>
> > **W2-(i): Why are different sampling ratios used for the two datasets?**
>
> The different sampling ratios were not intentionally chosen to impose different few-shot settings across datasets. Instead, we fixed the absolute number of training videos to 100 for both datasets when learning experience. Since XD-Violence and UCF-Crime have different training set sizes, this corresponds to 2.5% and 6%, respectively. In other words, what we controlled was the absolute annotation budget, rather than the relative sampling ratio. This design allows a more direct comparison of LRPO’s few-shot learning ability under the same labeling budget.
>
> > **W2-(ii): How are the few-shot subsets constructed?**
>
> The few-shot subsets are constructed by uniform random sampling across categories from each dataset’s training set, with a total of 100 videos selected for training. This is intended to reduce severe class imbalance in the sampled subset. We will make this construction protocol explicit in the revised version.
>
> > **W3: Construction of the shuffled variant.**
>
> The shuffled variant is constructed by preserving the same set of sampled frames and only randomly permuting their temporal order, rather than resampling a different set of frames. In other words, the ordered and shuffled inputs contain the same frame content and the same number of frames; the only difference is the frame order. This allows the performance difference between the two variants to more directly reflect the extent to which the model relies on temporal evidence for anomaly reasoning. The shuffling granularity is therefore at the frame level after sampling: for the same sampled sequence, we construct a temporally ordered version $\tilde{V}^{ord}$ and obtain the shuffled version $\tilde{V}^{shf}$ by applying a random permutation to that same sequence. We will state this construction more explicitly in the revised manuscript to improve reproducibility.
>
> > **W4: Writing and formatting issues.**
>
> We agree that there is still room to improve the writing and formatting of the manuscript. In particular, some symbol definitions and pipeline descriptions are relatively dense and may affect readability. In the revised version, we will further streamline the presentation, improve paragraph organization, and conduct a thorough check of notation and formatting throughout the paper. We will also correct minor issues such as "As illustrated in Figure. 2" to the proper form "Figure 2."

---

> > ### Author Rebuttal · Reviewer_DTt8 · 2026-04-01
> >
> > The authors have address all my concern.

---

> > > ### Author Response · Authors · 2026-04-02
> > >
> > > Thank you very much for the positive acknowledgement. We sincerely appreciate your time, careful reading, and supportive feedback.

---

### Official Review · Reviewer_4CJK · 2026-03-11

**Soundness:** 3
**Presentation:** 2
**Significance:** 3
**Originality:** 2
**Overall Recommendation:** 4
**Confidence:** 3

**Summary:**

This paper proposes a tuning-free anomaly detection framework that iteratively optimizes a textual anomaly experience pool instead of updating model parameters. By combining trajectory sampling, reward-guided reflection, and experience retrieval, the method improves video anomaly detection performance and sample efficiency across multiple benchmarks.

**Compliance With Llm Reviewing Policy:**

Affirmed.

**Key Questions For Authors:**

1. Can the authors better clarify what is fundamentally new in LRPO beyond iterative experience editing and retrieval-augmented prompting?

2. How sensitive is the method to the construction quality of scenario experiences and negative preference samples?

3. Can the authors provide a more controlled comparison under matched backbone / compute / annotation settings against the strongest tuning-free baselines?

**Limitations:**

The method still depends on strong external models and a multi-stage optimization pipeline, so its practical efficiency and simplicity may be more limited than the “tuning-free” framing suggests. In addition, the current evaluation does not fully establish whether the gains come from the proposed optimization principle itself or from richer textual experience augmentation.

**Strengths And Weaknesses:**

Strengths

* The paper addresses a meaningful setting: parameter-free adaptation for video anomaly detection.
* The overall framework is clear and reasonably well designed.
* Experiments are fairly comprehensive, with ablations on rewards, experience types, retrieval, and sample efficiency.

Weaknesses

* The methodological novelty is moderate: the main contribution is better framed as reward-guided verbalized experience editing rather than a fundamentally new policy optimization method.
* Some gains over prior tuning-free baselines are small, and the evidence for strong superiority is not fully convincing on all datasets.
* The reward design appears somewhat self-referential, since parts of the preference signal are derived from the same training data and generated experiences.

---

> ### Author Rebuttal · Authors · 2026-03-31
>
> We sincerely thank the reviewer for the positive evaluation of our work, including the meaningful problem setting (`parameter-free adaptation for video anomaly detection`), the overall method presentation (`the overall framework is clear and reasonably well designed`), and the experimental study (`experiments are fairly comprehensive`). We have carefully considered the reviewer’s constructive comments and provide detailed responses below. For clarity, we use "W#" to denote weakness comments, "Q#" to denote questions, and "L#" to denote limitations.
> > **W1/Q1: On novelty.**
>
> We agree that LRPO is not parameter-space RL; more precisely, it is inspired by relative policy optimization but instantiated in a **language-editable experience space**.
> Compared with prior iterative experience editing, LRPO differs in three aspects: (i) it optimizes a persistent and reusable **anomaly experience repository**, rather than revising prompts/output heuristics for individual cases; (ii) it relies on **group-wise multi-trajectory comparison** instead of single-output feedback; and (iii) it updates experience using **distilled relative semantic advantages**, rather than absolute output-level corrections.
> Compared with retrieval-augmented prompting, retrieved scenario experiences in LRPO are not generic context augmentation, but are used to calibrate anomaly decision boundaries, while the main transferable knowledge is stored in the optimized general experience. Importantly, Table 1 already shows that the gain is not from adding text alone: manual experience (68.47) and single-trajectory learned experience (66.78) are both clearly below LRPO-learned experience (73.17).
> > **W2/Q3: On controlled comparison.**
>
> We agree that the gains in the main table may appear modest, as those results are collected from original papers and are not fully aligned in backbone, temporal window, and implementation details. We therefore add a controlled comparison with VERA under its original backbone setting (InternVL2-8B) and the same weak-label setting, using both the 10s and 2s windows. **LRPO outperforms VERA under both settings**, with an even larger margin at 10s, indicating that the gain is not simply caused by temporal-window differences, but is better explained by LRPO’s optimization of linguistic anomaly reasoning experience.
> Method|Time Window (s)|XD-Violence (AP %)|UCF-Crime (AUC %)|UBNormal (AUC %)
> -|-|-|-|-
> VERA|10|70.11|86.55|-
> LRPO|10|73.93|87.23|-
> VERA|2|62.13|80.94|64.25
> LRPO|2|68.65|82.21|71.03
> > **W3/Q2: On self-reference and sensitivity.**
>
> We understand the concern that the preference reward may appear partially self-referential. However, it is not used in isolation:**LRPO is anchored by external weak labels** via the accuracy reward, and further constrained by the temporal dependency reward. In addition, the positive preference text is not free-form model self-feedback; it is constructed from the current sample under weak-label and rule-template constraints. Experimentally, **LRPO is also not highly brittle to generator quality**: using weaker models for scenario experience construction reduces AP only from 73.17 to 73.02 / 72.59, and replacing GPT-OSS-120B with GPT-OSS-20B for negative preference generation reduces AP only to 72.28. These drops are smaller than the gains brought by the corresponding components, suggesting that generator quality mainly affects the performance ceiling, while the main improvements come from LRPO’s core design.
> Scenario experience generator|InternVL3.5-8B|InternVL3.5-14B|InternVL3.5-38B
> -|-|-|-
> AP (%)|72.59|73.02|73.17
>
> Negative sample generator|GPT-OSS-20B|GPT-OSS-120B
> -|-|-
> AP (%)|72.28|73.17
> > **L1: On practicality.**
>
> We appreciate the reviewer’s concern about practicality. Although LRPO uses a three-stage pipeline, the **overall workflow remains lightweight in practice, with the full training process taking only about 2.3 hours**. The three stages serve complementary roles in experience optimization: (i) trajectory sampling explores the anomaly preference space, (ii) semantic advantage extraction converts reward differences into actionable signals, and (iii) experience optimization consolidates them into reusable experience. Moreover, LRPO does not rely on the strongest external model to remain effective, as replacing GPT-OSS-120B with the smaller InternLM3-8B leads to only a 0.4% drop (Table 3).
> > **L2: On Optimization Effect.**
>
> We appreciate this concern and agree that the current evaluation should better separate the effect of optimization from that of richer textual experience augmentation. To this end, we keep the experience injection format fixed and examine performance as the experience is progressively refined during LRPO. AP improves steadily from 67.63 to 73.17, suggesting that **the gain comes not only from adding experience, but also from continually improving its quality through optimization**.
> Experience Stage|Initial|1/3|2/3|Final
> -|-|-|-|-
> AP (%)|67.63|70.04|72.96|73.17

---

> > ### Author Rebuttal · Reviewer_4CJK · 2026-04-04
> >
> > Thanks to the authors. The rebuttal addressed my main concerns.

---

> > > ### Author Response · Authors · 2026-04-04
> > >
> > > We truly appreciate the reviewer’s encouraging response. It is reassuring to know that our clarification resolved the key concerns, and we are grateful for your thoughtful evaluation and consideration.

---

### Decision · Program_Chairs · 2026-04-30

**Decision:**

Accept (regular)

**Comment:**

Reviewers were mainly concerned about the exact novelty in relation to prior experience with editing or retrieval-based prompts, the fairness and control of baseline comparisons, the sensitivity and potential self-referential nature of reward design, as well as several issues regarding clarity and repeatability, such as editing mechanisms, few - shot subset construction, temporal obfuscation protocols, notation, and presentation. In the rebuttal, the authors addressed these concerns in a concrete and persuasive manner. They clarified the methodological distinctions of LRPO, added controlled matching - backbone comparisons, provided evidence that the method is not overly susceptible to generator selection, explained the effects of optimization beyond simple text augmentation, and provided the missing implementation details and dataset statistics. Additionally, all reviewers explicitly responded that their concerns had been fully resolved after the refutation, and the final discussion indicated unanimous approval.